# Season, wind speed, and seasonal rain are major drivers of a regional aeolian sediment transport model

Andrew Kulmatiski[1]*, Mehmet Ozturk[1], Kelvyn K. Bladen[2], Janice Brahney[3], Michael C. Duniway[4]

1 Wildland Resources, Utah State University, Logan, Utah, United States of America, 2 Department of Mathematics and Statistics, Utah State University, Logan, Utah, United States of America, 3 Watershed Sciences, Utah State University, Logan, Utah, United States of America, 4 U.S. Geological Survey, Southwest Biological Science Center, Moab, Utah, United States of America

* andrew.kulmatiski@usu.edu

## Abstract

Wind erosion and sediment transport continue to increase in many parts of the world, leading to decreased soil quality, accelerated snow-melt, respiratory diseases, and traffic accidents. The processes that control sediment transport are well understood at small scales of mm to m but are less well understood at larger scales of km to hundreds of km. Here we test four approaches aimed at improving the variance explained in sediment transport measured in a network of 52 horizontal sediment flux collecting devices located on the Colorado Plateau, USA. First, switching from a regression tree to random forest statistical analysis increased the variance in sediment transport explained from 58% to 91%. Soil moisture as a single variable explained 52% of variation in sediment flux, but had a negligible effect on a random forest model with season (Winter, Spring, Summer), wind speed, and seasonal total precipitation. Similarly, adding four years of new data to an existing five-year dataset or adding measurements of soil roughness and grazing failed to improve variance explained. By explaining 91% of the variance in sediment transport, our model pro-vides baseline model for understanding sediment transport on the landscape scale. Dust flux networks in new regions would likely need to collect at least 300-500 sam-ples to describe variation in sediment transport values using random forest analyses of the effects of season, wind speed, seasonal rain and vegetation type.

## Introduction

Sediment transported by wind erosion decreases soil stability and productivity, and plant and biocrust productivity, and increases nutrient loading and snowmelt in alpine systems [1–4]. Wind erosion is also a public health concern because it can cause human respiratory problems including coccidiomycosis and silicosis as well as traffic

**Data availability statement:** All data used in this study are available as appendices or in the following data publication... Tyree, G.L., L. A. Zeller, J. Belnap, E. L. Geiger, T. W. Nauman, and M. C. Duniway. 2024, "Dust mass and horizontal aeolian sediment flux data from a sampler network on the Colorado Plateau, USA." U.S. Geological Survey data release. https://doi.org/10.5066/P1ZHQX9W.

**Funding:** This research was supported by the Utah Agricultural Experiment Station, Utah State University, and approved as journal paper number 9854. The Ministry of National Education and the Ministry of Agriculture and Forest of Turkey supported M. Ozturk.

**Competing interests:** The authors have declared that no competing interests exist.

accidents [5–7]. Aeolian research, designed to understand the factors that determine wind erosion, dust transport, and inform land management has been performed for nearly a century [8–10]. This long history of research has effectively described the mechanisms of wind erosion at small scales of mm to meters [11,12]. Broadly, wind erosion can be described as a balance of forces that remove particles from the surface (i.e., drag or lift associated with wind), and forces that oppose particle removal (e.g., capillary or chemical binding forces associated with soil moisture and organic material, etc.) [13–15].

Despite a good understanding of factors affecting wind erosion at small spatial scales, there is a recognized gap in knowledge of the factors that determine wind erosion at larger spatial scales of km to hundreds of km [5,10,16,17]. At large scales, it is thought that horizontal sediment mass will be a function of many of the same factors that are important at small scales, though likely to different degrees [7,18]. A primary problem for testing these predictions is that large-scale measurements of ground-level sediment transport are needed but uncommon [11,19,20]. Further, because many factors covary or interact, it is difficult to measure sediment transport under all potential combinations of factors [10,21]. For example, soil organic matter facilitates water infiltration, increases water-holding capacity, and improves aggregate stability while the amount of soil organic matter is affected by vegetation type and amount [7,22]. Therefore, modeling approaches that can incorporate correlated data streams are needed [23].

A series of studies using a large network of sediment collectors placed on the Colorado Plateau, USA have provided insight into the factors that drive sediment flux at large scales. Similar to small-scale studies, these large-scale studies found that wind speed, temperature, soil type, and vegetation cover were important for determining sediment transport particularly in relatively undisturbed soils [23–25]. In contrast to small-scale studies, these studies found less support for the role of seasonal or annual precipitation on sediment flux [23,25]. It is possible that seasonal and annual precipitation is too coarse of a variable and that information about soil moisture that is temporally linked to wind speed may be better correlated with sediment transport since it has a direct impact on wind erosion threshold [13,15]. Soil moisture, therefore, is known to be important at small scales and is likely to be important at large scales, but we are not aware of a study that has assessed the effect of soil moisture on sediment transport at large scales. Here, we address this knowledge gap by including estimates of soil moisture as a predictor of sediment flux in regional sediment flux measurement network.

In addition to testing for soil moisture effects, we also tested three additional approaches to improve understanding of the variation in sediment transport using a subset of the network of the rangeland sediment transport collectors reported by Flagg et al. (2014) and Nauman et al. (2018, 2023). First, we tested model improvement with four years of new data because we anticipated that it would require a large amount of data to effectively describe a large number of interactions that can occur among many different and varying parameters. Next, we used a random forest model to describe sediment transport. The random forest modeling approach is well suited

to incorporating correlated data streams and identifying non-linear relationships among variables and has been shown to improve upon regression tree analyses [26,27]. Third, we added measures of soil roughness and grazing intensity because these variables have been found to be associated with sediment transport [24,25,28,29].

## Materials and methods

### Study area

The study area and design of the rangeland sampler network are described in Flagg et al. (2014) and Nauman et al. (2018, 2023). Briefly, the study area (Fig 1) covers a roughly 6270 km$^2$ area of southeastern Utah, and extreme western Colorado on the Colorado Plateau between 1,000 and 2,200 m. The area has a dry climate characterized by hot summers and cold winters. Mean annual precipitation and temperature range from 150 mm to 400 mm and from 9°C to 15°C, respectively. Dominant plant types (> 25% total cover) are blackbrush and ephedra shrublands (*Coleogyne ramosissima* and *Ephedra viridis*: Blackbrush), exotic annual plant communities that occurred most commonly on Mancos shale-derived soils (*Bromus tectorum, Halogeton glomeratus, Salsola tragus*: Mancos), perennial grasslands (*Bouteloua gracilis* and *Hilaria jamesii*: Grassland), Pinyon-Juniper woodlands (*Pinus edulis* and *Juniperus osteosperma*: PJ), sagebrush shrublands (*Artemisia tridentata*: Sagebrush) and saltbrush shrublands (*Atriplex confertifolia, A. corrugata*, and *A. gardneri*: Saltbrush). Land cover groupings were established in Flagg et al. (2014) and reported in Flagg et al. (2014) and Naumann et al. (2023). Soils are predominantly derived from either sandstone or Mancos shale though siltstone and limestone are also present. Sites with soils derived from Mancos shale typically have a strong physical crust formed by cementation of soluble salts and clays and are poor for plant growth and were classified by their lack of perennial vegetation (i.e., sites where perennial plants represent < 20% total cover; [30].

### Sediment transport monitoring

The U.S. Geological Survey (USGS) started a landscape-scale study to monitor sediment horizontal mass transport ($q$) on the Colorado Plateau in 2004 by installing 85 passive horizontal sediment transport collectors [Big Spring Number Eight (BSNE) sediment samplers; [31,32]. Some of the BSNE collectors failed or were removed during the sampling period, leaving 52 samplers that were collected throughout the study. BSNEs were used to collect sediment mass at 15, 50, and 100 cm above the soil surface, though data from 15 cm are not reported because vegetation often obstructed

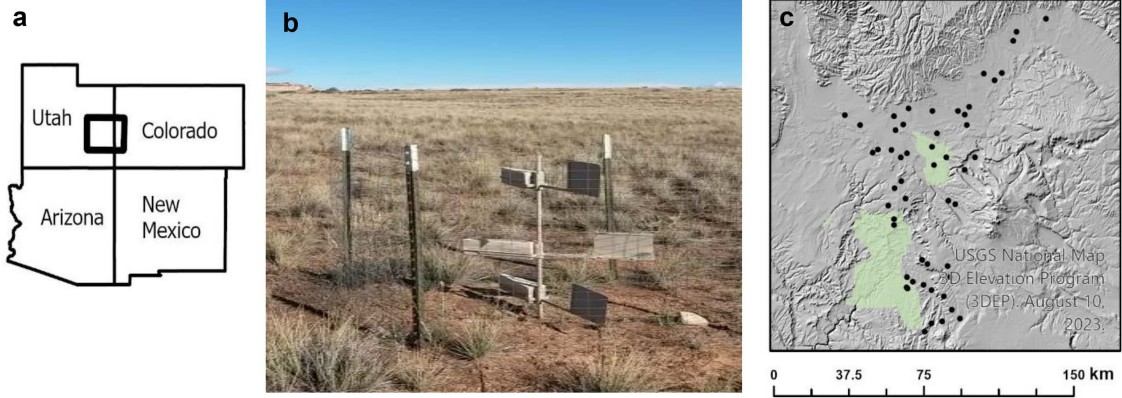

**Fig 1. Study area (a), location of 52 sediment collection devices (b) and photograph of one of the collection devices in the field, western USA.** The 'big spring number eight' (BSNE) sediment collectors have a wind vein to orient an opening into the wind at 15, 50, and 100 cm heights. **(b)** Thick box crossing the Utah/Colorado border shows the extent of the study shown in panel c. Study site locations shown as black dots. Green polygons indicate Canyonlands (left) and Arches (right) national parks.

the movement of these collectors (Fig 1; [23]). Each BSNE was installed 50 m from unpaved roads and at least 1 m from large vegetation obstructions. Sample collections were performed before the samplers were filled to capacity, three times a year in early spring (Feb-Mar), mid-summer (~Jul), and fall (Oct-Nov). Sediment data from 2004 to 2012 were described using regression tree analyses by Flagg et al. (2014), from 2004 to 2015 and compared to wind erosion from roads and off-highway vehicle area by Nauman et al. (2018; 100 cm flux only), and from 2017 to 2020 and compared to flux from reclaimed oil and gas pads by Nauman et al. (2023; 100 cm flux only). Consistent with Flagg et al. (2014), here we analyze the sum of sediment transport from 50 and 100 cm heights. In addition, we analyze 11 years of continuous data, adding four years of data (2015–2018) to the Flagg et al. (2014) dataset and adding the 50 cm height flux to the data from Nauman et al. (2018) [33]. This study involved collecting dust samples from an established network of dust collectors placed on public lands. Required permits for establishing and maintaining these study sites on public lands were obtained from the Bureau of Land Management and National Park Service (see Acknowledgements).

Sediment collected in the boxes of the BSNEs was washed using deionized or purified water into plastic bags and dried in an oven at 60°C to constant mass (0.0001 g). Organic litter >1 mm in diameter and/or longer than 1 cm in length and dead insects were weighed separately. Sediment-transport ($q$) was calculated by dividing the recorded sample mass by the area of the BSNE opening (10 cm$^2$) and the sampling period duration at each collection height (g meter$^{-2}$ day$^{-1}$). Values from the 50 cm height and the 100 cm height collectors were summed. The horizontal mass flux data are available from the US Geological Survey Sciencebase Catalog [33].

## Site conditions

Soil texture, soil stability, biological soil crust cover, annual plant cover, percent perennial plant cover, and plant gap means for each of the 52 collections sites are reported in Flagg et al. (2014). Ground cover and canopy gap size were re-measured April 2017 for the 2015–2018 data. Fifty-meter transects in the directions of 110, 220, and 330 azimuths from true north were located around BSNE samplers [23]. Plant and soil cover was measured using line-point intercept hits in every 25 cm, recorded as biological soil crusts, rock fragments, woody debris, plant litter, bare soil, and plant species [34]. Canopy and basal gaps were also measured and classified into four groups as 25–50 cm, 50–100 cm, 100–200 cm, and larger than 200 cm [34]. Further, two new site conditions were measured. Soil surface roughness (mm) was calculated using the chain method [35]. A 40-mm chain was placed on the soil surface every 5 m (9 points per transect), and the straight-line length (parallel to the soil surface) of the chain was measured three times (in the angle of 110$^0$, 220$^0$, 330$^0$) at each point. Second, we used a qualitative assessment of grazing condition (grazed, no graze, some graze) for each BSNE site reported in [25] (https://doi.org/10.15482/USDA.ADC/1528278).

## Meteorological data

Precipitation and hourly wind speed data were obtained from the Moab, Canyonlands meteorological station (KCNY; [36]), which is located near the center of the study area. Wind data was calculated as total hours of the wind exceeding 4, 8, and 12 m s$^{-1}$ as well as mean hourly wind speed (wind) and mean gust wind speed. Mean climate data was calculated for dates between each sample collection.

## Soil moisture modeling

Volumetric soil water content was simulated for each BSNE site using the soil water movement model, Hydrus [37,38]1D. Hydrus simulates water and energy transport as a function of meteorological data, including precipitation, temperature, radiation, wind speed, and relative humidity and soil physical parameters. Daily precipitation and temperature data were taken from the 4 km resolution PRISM dataset (http://www.prism.oregonstate.edu/recent/). Soil physical properties were estimated using neural network predictions (Rosetta Lite Dynamic Linked Libraray [39]] based on measurements of soil texture (0–10 cm) reported by Flagg et al. (2014). Evapotranspiration was estimated using the Penman-Monteith equation

[37,38]. The hydraulic sub-model was a van Genuchten-Mualem model with no hysteresis. Neural network predictions of soil hydraulic parameters were made using soil texture and bulk density data. The water flow boundary conditions allowed for surface runoff and free drainage. Plant height was assigned a value of 50 cm and leaf area index was assigned a value of 1 which are consistent with personal observations and similar to previous observation for nearby systems [40,41]. Hydrus 1D model simulations were calibrated using volumetric soil water content data from three Natural Resources Conservation Service sites (https://wcc.sc.egov.usda.gov/nwcc/site?sitenum=461, 572, 2131), and two USGS stations ('Dugout Ranch' and 'Needles'; CLIM-MET data obtained from http://esp.cr.usgs.gov/info/sw/clim-met/; [42]). Model estimates of volumetric soil moisture at 5, 15 and 30 cm depths for each site were then used as a predictive variable of sediment transport.

## Data analyses

We primarily rely on random forest analyses (described below) because random forest has been found to be a powerful approach for describing variation in a dataset when multiple covarying explanatory variables are available [26,27]. However, because random forest analyses complicated and difficult-to-interpret models, we also include univariate tests for differences in sediment transport associated with land cover (Blackbrush, Grassland, Mancos, Pinyon/Juniper, Sagebrush, Saltbrush), grazing condition (No graze, Some graze, Grazed), and season. Univariate tests were used because they describe easy-to-interpret relationships between two variables. The effects of these variables were tested using univariate tests because their of their importance to management and interpretability. For these tests, we used completely randomized linear mixed models with land cover, grazing condition, or season as the fixed effects. Sediment transport values were log-transformed as necessary to meet assumptions of normality and homogeneity of variance. Means were compared using the Tukey-Kramer adjustment for multiple comparisons at the *alpha* = 0.05 level. Tests were performed using Proc Glimmix in the SAS programming language (SAS v. 9.4).

We used a random forest model (RF; [43] to describe the relationship among many interacting site variables (Tables 1 and 2) and sediment transport and to build a predictive model of sediment transport. RF creates many classification or regression trees and provides the mode or mean of these trees. RF corrects for the tendency for decision trees to overfit data [26,27,44]. The proportion of variation of the response variable (i.e., sediment transport) explained by the tree (variation explained) determines the quality of a regression tree or a RF regression tree [45]. Therefore, the quality of regression tree and RF model results was assessed by comparing the variance explained by each model. Similarly, the ability of soil moisture data and a larger dataset to explain sediment transport was assessed by comparing the variance explained with each addition to the dataset. To estimate variable importance, values for each variable were randomized, one at a time, and the prediction error in models with randomized and non-randomized values calculated [26]. Large prediction error indicates large variable importance (VIMP). RF Analyses were performed in R environment using the randomForestSRC package [46].

We used RF to perform several analyses. First, we tested the ability of RF to explain variation in sediment transport relative to a previously used regression tree modeling approach [23]. To do this, we compared the variation in sediment transport explained by regression tree and RF models (Table 2) [23]. Second, we tested the ability of more data to improve variation explained. To do this, we compared RF model results from the 2009 to 2013 dataset (951 observations) to RF model results from the 2009 to 2018 dataset (1458 observations). Third, we tested the ability of soil moisture at 5, 15, and 30 cm to explain variation in sediment flux. To do this, we tested the ability of soil moisture data alone to predict sediment transport, then we tested the ability of soil moisture data to improve predictions from the previously described model. Finally, we added grazing condition and soil roughness data.

## Results

### Sediment transport

Sediment transport (sum of 50 and 100 cm measurements) ranged from 0.00 to 503.33 g m$^{-2}$ day$^{-1}$ with a mean of 6.96 ± 26.09 g m$^{-2}$ day$^{-1}$ (mean ± SD). Sediment transport was smallest in PJ cover and greatest in Blackbrush (Fig 2; $F_{5, 1459}$

**Table 1. Variables used to describe variation in sediment transport.**

| Variable name | Description | Units |
|---|---|---|
| Aggstab | Aggregate stability (1–5) | Category |
| Avggust | Average of the highest 3 second wind speeds from each hour | M sec$^{-1}$ |
| B25, B50, B100, B200, B300 | Sum of gaps between plant stems where gap distances are 25, 50, 100, 200, or 300 cm, respectively | Cm |
| Bareground | Percent of ground without litter or vegetation cover | % |
| Bsc | Biological soil crust | % |
| C25, C50, C100, C200 | Sum of gaps between plant canopies where gap distances are 25, 50, 100, 200, or 300 cm, respectively | Cm |
| Condition | No graze, some graze, heavy graze | Category |
| Mois5, Mois15, Mois30 | Volumetric soil moisture, at 5, 15, or 30 cm | cc water cc$^{-1}$ soil |
| Roughness | Straight-line length of a 40 mm chain lain on surface | mm |
| Season | Winter, Spring, or Summer | Category |
| Seasrain | Sum of rain during the sampling season | Mm |
| Sand | Percent sand in soils | % |
| Vegetation | Dominant surface cover (Blackbrush, Grassland, Mancos, PJ, Sagebrush, Saltbush) | Category |
| Wgust | Average of maximum 3 sec wind speed in an hour | M sec$^{-1}$ |
| Wind | Average wind speed per hour during sampling period | M sec$^{-1}$ |
| Wind4, Wind8, Wind12 | Sum of hourly wind speeds above 4, 8, or 12 M sec$^{-1}$ | hours |
| Year | Year of sampling | Year |

**Table 2. Model summary and variance in sediment transport explained by each model.**

| Model name | Model description | Variance explained |
|---|---|---|
| Flagg | Regression tree analysis of 2009–2018 data using wind, vegetation type, season, precipitation, soil stability, basal gap size, canopy gap size, % sand, and % soil crust, soil moisture | 58% |
| RF1 | Random forest of Flagg dataset (no soil moisture) | 91% |
| Moisture | Random forest of 2009–2018 data using only soil moisture | 52% |
| RFMoist | RF1 with soil moisture data included | 92% |
| Grazing and Roughness | RF model of 2009–2018 data using only grazing and soil roughness | 9% |
| RF Full | RFMoist with Grazing and Roughness added | 92% |
| RF Full short | RF Full for 2009–2014 data | 91% |

= 5.16, p < 0.001). Sediment transport was the smallest in winter and greatest in spring (Fig 3; $F_{2, 1459}$ = 293, p < 0.0001) and smaller in No Graze and Some Graze than in Grazed (Fig 4; $F_{5, 1462}$ = 5.34, p = 0.005).

### Random forest and regression tree models

Whether parameterized with data from 2009-2014 or 2009–2018, the RF model explained 91% of variance in sediment transport (Table 2). In contrast to RF, when the full dataset was analyzed using regression tree analysis, 58% of variance was explained. Soil moisture alone explained 52% variation in sediment flux in the 2009–2018 dataset, however, adding soil moisture to other variables (e.g., season, wind speed, etc.) had a trivial effect on the model, increasing variance explained to 92%. Grazing and soil roughness alone explained only 9% of variance in the 2009–2018 dataset. Adding grazing and soil roughness variables to the other variables did not improve the model.

A variable importance plot revealed that 'season' (Winter, Spring, Summer), 'wind' (average wind speed; m sec$^{-1}$), and 'seasonrain' (sum of rain in the season; mm) explained the most variation in the RF model (Fig 5). Given that most of the variables appeared to contribute much less to variance explained, we created a simpler model with the most important

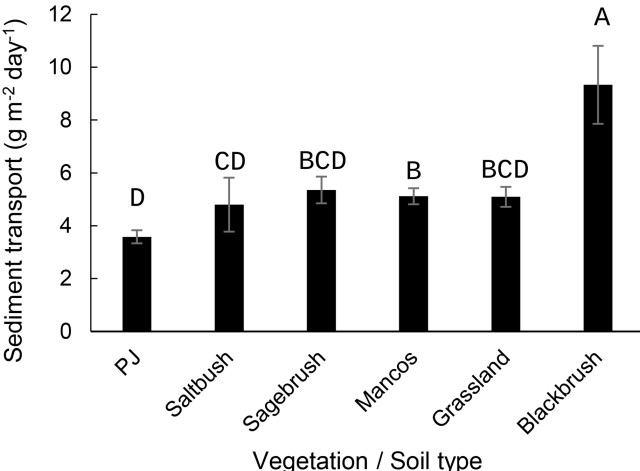

**Fig 2. Mean daily sediment transport (g m$^{-2}$ day$^{-1}$) under different vegetation types and soil cover in southeastern Utah, USA, 2004 to 2018.** Sediment transport was greatest in the Blackbrush vegetation type and less in PJ than in Mancos. Saltbush was also less than Mancos. Columns with a different letter are significantly different at the alpha = 0.05 level in a *Tukey's HSD post-hoc test. PJ = piñon/ juniper.*

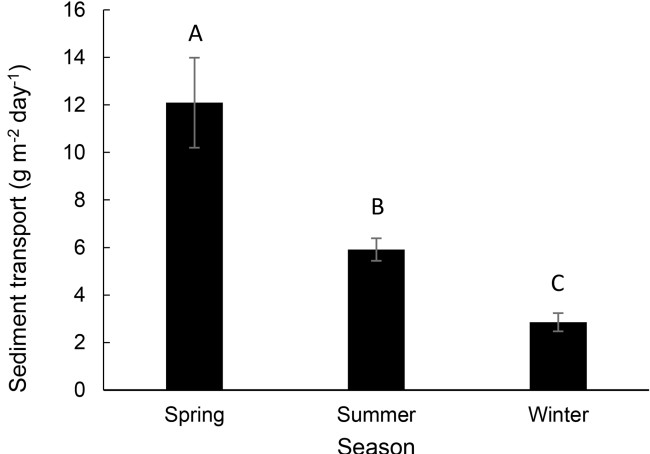

**Fig 3. Mean sediment transport (g m$^{-2}$ day$^{-1}$) by season (Spring, Summer, Winter).** Sediment transport was greatest in Spring, less in Summer and least in Winter.

variables (season, wind speed, seasonal rain, and wind speed above 8 and 12 m sec$^{-1}$). This simplified model explained 62% of variance in the sediment transport dataset (not shown).

To explore the relationship between important variables and sediment transport, we examined the partial dependence plots. Partial dependence plots show the relationship between a variable of interest and the random forest model output and can show linear or non-linear relationships [26]. Partial dependence plots show the log sum values which are the relative logit contribution of the variable on the class probability from the perspective of the model. Larger log sum values mean that the larger values of the predicted variable (i.e., sediment flux) are more likely in the model [27]. For the top variable, 'season', the log sum of predictions was 0.89 and 0.87 for Spring and Summer and 0.68 for Winter, indicating that Spring and Summer were associated with large sediment transport relative to Winter. The top continuous variables in the variable importance plot were 'wind', 'wind4', 'wind8', 'wind12','avggust', 'seasonrain', and soil moisture at 5, 15, and

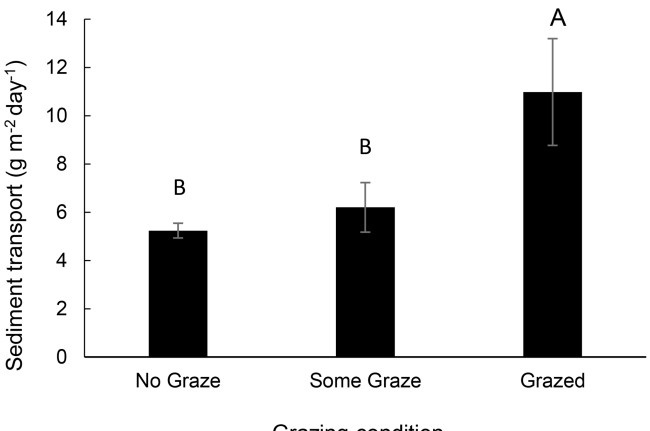

**Fig 4. Mean sediment transport (g m⁻² day⁻¹) by grazing condition (Grazed, Some Graze, and No Graze).** Grazed sites realized more sediment transport than 'No Graze' or 'Some Graze' sites.

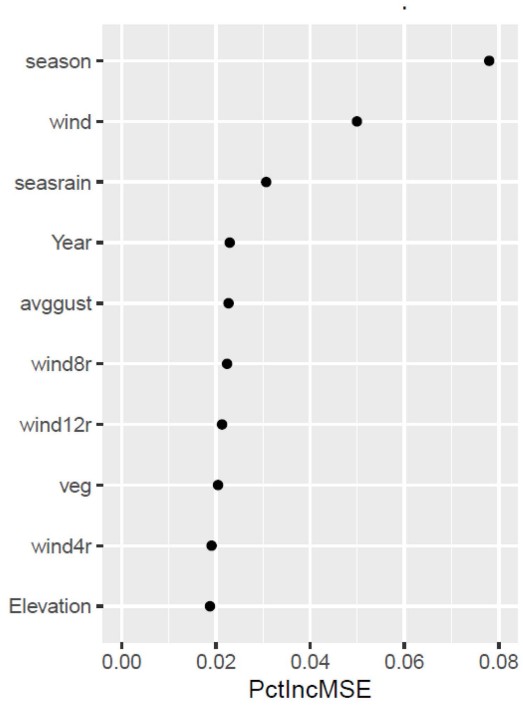

**Fig 5. Variable importance in Random Forest model of seasonal sediment transport with 1468 observations from 2009 to 2018.** Variable importance was assessed by the percent increase in mean squared error when a variable was removed from the model (PctIncMSE). Small values of PctIncMSE indicate that removing a variable had a small effect on the quality of the model. The ten most important variables in the model were season (winter, spring, summer), wind speed, seasonal rain (seasrain), year, average wind gust speed (avggust), hours of wind above 8 m sec⁻¹ (wind8r), hours of wind above 12 m sec⁻¹(wind12r), ground cover types (veg), hours of wind above 4 m sec⁻¹ (wind4r), and elevation.

30 cm (Fig 6). Sediment flux generally increased with average wind speed, but showed a marked increase at ~2.4 m sec⁻¹. Sediment transport similarly increased with the hours of wind speeds above 8 and 12 m sec⁻¹, though these effects showed a more consistent increase until 300 hours at 8 m sec⁻¹ and 55 hours at 12 m sec⁻¹. Sediment flux generally

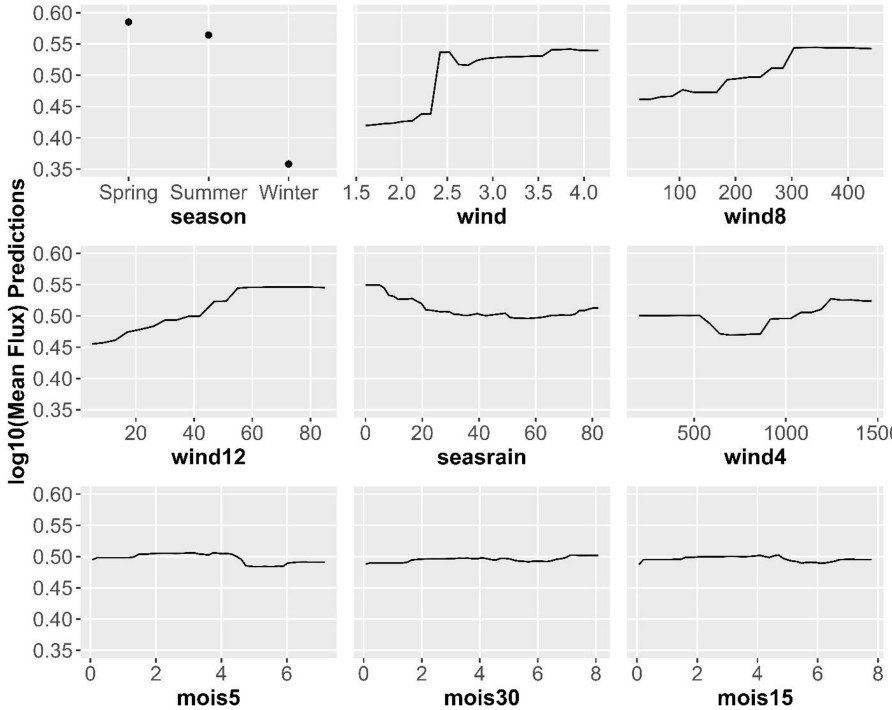

**Fig 6. Partial dependence plots.** Plots show the relationship between variation in a single variable (x-axis) and the relative logit contribution of the variable on the class probability from the perspective of the model. In other words, large values (in the y-axis) mean that large values of sediment transport are more likely in the model.Values for peak wind, for example, show that sediment transport increases with wind speed, particularly when peak wind speeds exceed roughly 4 m s$^{-1}$. Note that axis values are not the same among variables.

decreased with increasing seasonal rain, though most of this effect occurred between 0 and 30 mm precipitation. Sediment flux was generally greater when 5 cm soil moisture was < 5%. Interestingly, sediment flux increased as soil moisture at 30 cm increased. The relationship between sediment flux and soil moisture at 15 cm appeared weak and idiosyncratic.

## Discussion

This study improved upon previous analyses of a regional dust flux dataset by 1) testing the effects of soil moisture, 2) increasing dataset size, 3) adding soil roughness measurements, and by using a random forest (RF) modeling approach, which was previously found to improve upon previous regression tree modeling approaches. By using a random forest (RF) model, increasing sample size, simulating soil moisture, and testing grazing and soil roughness effects, this research improved the explanation of variance in sediment transport from 58% to 92%. The vast majority of this improvement was caused by switching from a previously-used regression tree model to a RF model. The RF model improved the variance explained from 58 to 91%. Soil moisture alone explained 52% of variance, but caused a trivial increase in variance explained (91% to 92%). Similarly, adding four years of data (517 observations) to a five-year dataset (951 observations) or adding grazing condition or soil roughness information failed to improve variance explained.

Our final dataset and analysis provided a high-quality model of sediment transport on the Colorado Plateau that provides a foundation for understanding sediment transport in other regions based on weather and site conditions. Our results suggest that sediment transport sampling networks established in new regions should use RF models and include at a minimum measures of wind speed, precipitation, and vegetation type. A simple RF model with six precipitation and wind speed variables explained 62% of variation in sediment transport. Including other variables such as soil moisture,

elevation, grazing, soil texture, soil roughness, and vegetation gaps will provide incremental improvements on model predictions.

The RF modeling approach resulted in an impressive improvement in model performance. This is consistent with previous applications of this approach [27,44]. Implementation of the RF approach is similar to the regression tree approach so the RF approach is expected to provide superior performance in future analyses. The RF model created in this study can be used to simulate sediment transport under various conditions [26]. It is possible, for example, to use the model created here to provide quantitative predictions of sediment transport with anticipated climate conditions (i.e., with climate change), changes in grazing intensity or changes in vegetation type [26,47]. Whether or not the model can explain sediment flux in new landscapes remains to be determined and would require testing with new sediment flux data in new areas.

Increasing the sediment transport sample size from 951 observations over five years to 1468 observations over nine years had little effect on the variance explained. To further explore the effect of data availability on model quality, we randomly selected 100, 300, 600, and 1200-point datasets. This analysis suggested that variance explained increased rapidly until roughly 300 samples were available for analysis (Fig 7). In our dataset this represented roughly 50 sites collected three times a year for two years. Sample designs in new areas will be a function of landscape heterogeneity and interannual weather variability, but new sediment transport networks should plan to collect at least 300–500 samples.

Consistent with previous research, our model indicated that wind speed is the primary determinant of sediment transport. The large effect of wind speed helped explain that sediment transport was greatest in the spring because wind speeds were greatest in the spring (3.7 m sec$^{-1}$) and less in the summer (2.8 m sec$^{-1}$) and winter (2.0 m sec$^{-1}$). Also consistent with other studies [48–50], though not investigated with the previous studies at the study site [23–25], we found a relationship between sediment transport and soil moisture. We expected that the addition of soil moisture estimates would greatly improve model predictions because soil moisture has been shown to decrease sediment transport at small scales. Consistent with this expectation, a model using only soil moisture predicted 52% of the variance in sediment transport. This occurred even though soil moisture is generally high in winter when sediment flux is low and high in the spring when sediment flux is high. This suggests that soil moisture is important for controlling entrainment and transport of sediment through adhesion and not because soil moisture is correlated with season [51,52]. Typically, in wind erosion studies, only soil moisture in the top few cm are considered [13,15,32,53], but here we tested for the effects of soil moisture at 5, 15 and 30 cm. Consistent with theory, sediment flux decreased as soil moisture increased at 5 cm. However, we observed the opposite pattern with soil moisture at 30 cm: sediment flux increased with soil moisture at 30 cm (Fig 6). It is not clear why

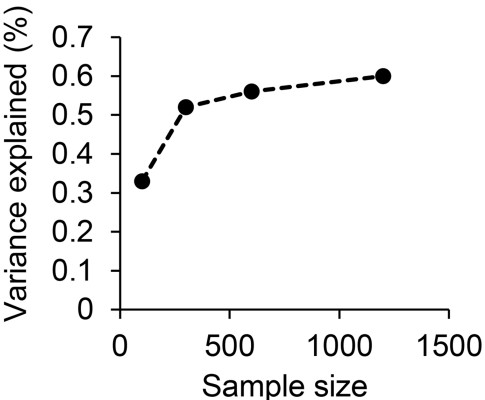

**Fig 7. The variance explained in sediment transport increased with sample size, but sampling beyond roughly 500 data points provided diminishing improvements.**

this occurred, though several explanations are possible. Deeper soil water may be associated with wetter ecosystems with greater productivity and less sediment transport. Alternatively, it is possible that shallow soil moisture was associated with smaller and more localized precipitation events, whereas deeper soil moisture was associated with more widespread precipitation events that had larger-scale effects on sediment transport. Perhaps most likely, it is possible that fast spring winds are associated with deeper soil percolation [54].

Even though soil moisture was an important variable on its own, it explained little variance in our full random forest model. We expected daily, depth-specific soil moisture estimates would explain more variance than the observed 1% increase in variation explained in the RF model [48,49]. The model we were trying to improve upon included information about seasonal precipitation. It appears that seasonal precipitation data explained much of the variation explained by soil moisture data. Consequently, where it is not possible to collect or simulate soil moisture, precipitation data provides a reasonable parameter for understanding soil moisture effects on sediment transport [24].

Vegetation types are believed to be important because plant structure changes aerodynamic roughness [14], but our results suggested more complicated effects of vegetation at the landscape scale. Sediment transport was highest in black-brush communities, yet blackbrush communities have a somewhat similar structure as sagebrush and saltbush. Mancos sites, which are dominated by saltbush, have the lowest stature and largest canopy gaps [23] but did not demonstrate the highest sediment transport. Broadly, while vegetation effects were observed, these effects were not consistent with plant structure. It is likely that sediment transport was greatest in blackbrush communities because blackbrush abundance was positively correlated with sand content [23]. So, while plant structure may play a role, other interacting factors appeared to determine sediment transport associated with vegetation types.

Results suggest that sediment transport is likely to increase with climate change in the future. Wind speed was a primary determinant of sediment transport and wind speed is likely to continue to increase with temperature in the future [55,56], so sediment transport is also likely to increase in the future. Further, Munson et al. [57] reported decreases in vegetation cover on the Colorado Plateau with warming. Decreased plant cover is likely to increase sediment transport because decreased plant cover is associated with greater exposed soil which increases sediment transport [57,58], particularly in disturbed sites [24]. Finally, where increasing temperatures decrease soil moisture, this effect will also increase sediment transport.

Vegetation and soil cover is one of the few factors that can be managed to mitigate anticipated increases in sediment transport. For example, there may be potential to reduce sediment transport in the blackbrush communities by promoting inter-shrub stability through restoring biological soil crusts and perennial grasses – both of which demonstrated less sediment transport, especially in sandy soils [7,21,57]. However, restoration of degraded rangelands on the Colorado Plateau has been shown to be challenging, especially biological soil crusts [59,60]. Thus, limiting soil disturbance of these wind erosion prone soil-vegetation settings maybe the best management approach to reduce accelerated wind erosion [7]. Our results, similar to those of Nauman et al. [24], suggest that light or no grazing that protects soil crusts or management that increases or maintains biological soil crust will help decrease sediment transport, particularly in blackbrush communities. Grazing, as a single variable, was associated with greater sediment transport (Fig 4), but grazing condition was not a top variable in our models. This occurred because grazed conditions were associated with the shale land cover type which provided a better explanation for sediment transport across the dataset. Grazing may be more important in other systems where grazing intensity is greater or grazing effects are not as well captured by other variables. Our results are consistent with findings demonstrating that biological soil crust can increase the infiltration rate, soil moisture content, and limit initiation of wind erosion [21,61], which can then in turn support vegetation communities that provide further protective cover from erosive winds [7].

Results presented here inform our current understanding of wind erosion processes in drylands at a landscape scale, particularly in relatively undisturbed soils. Relatively undisturbed soils dominate the landscape, so our results provide valuable insight into landscape-scale patterns. However, it is important to note that sediment flux can be much greater on disturbed

soils, including the nearly doubled sediment transport seen here with higher grazing activity and orders of magnitude increases associated with roads, off-highway vehicles, wildfire, and other disturbances [7,25]. As a result, if disturbed soils represent more than a few percent of the landscape, they are likely to have large effects on regional sediment flux [25].

Use of a random forest model improved the variance in sediment transport explained relative to regression tree analysis and is expected to provide superior analyses of new datasets in other systems. Increasing sample sizes beyond 300–500 samples provided little additional explanatory power in describing landscape patterns. As a result, moving dust collector networks after a few years to better capture landscape heterogeneity would likely provide more explanatory power than collecting more data from the same sites across years. Soil moisture data provided little additional improvement beyond other factors, but was important on its own. By explaining 92% of variation in observed sediment flux, the model developed here provides a strong foundation for understanding sediment flux across the landscape, in response to climate change, and in new regions.

## Supporting information

**S1. R code for random forest analyses.**
(PDF)

**S2. SAS code for univariate analyses.**
(DOCX)

## Acknowledgments

We would like to thank Erika Geiger and Hilda Smith for their help facilitating the USU field sampling, Jayne Belnap for her part in establishing the sampling network, Travis Nauman for his work curating the data, and the many other staff at USGS that maintained the network of samplers over the decades. For sites on federal land, we obtained the necessary permits (Bureau of Land Management permit numbers: UT-Y010-2012–0100, UTU-82446, UTU-87617, UTU-87666, UTU-75486; National Park Service study numbers: CANY-00013, CANY-00047, CANY-00048, CANY-00124, CANY-00126, ARCH-00078, ARCH-00082, ARCH-00008, ARCH-00046). Any use of trade, firm or product names is for descriptive purposes only and does not imply endorsement by the U.S. Government.

## Author contributions

**Conceptualization:** Michael C. Duniway.

**Data curation:** Mehmet Ozturk, Michael C. Duniway.

**Formal analysis:** Andrew Kulmatiski, Mehmet Ozturk, Kelvyn K. Bladen.

**Funding acquisition:** Andrew Kulmatiski, Mehmet Ozturk.

**Investigation:** Andrew Kulmatiski, Mehmet Ozturk, Janice Brahney, Michael C. Duniway.

**Methodology:** Andrew Kulmatiski, Mehmet Ozturk, Michael C. Duniway.

**Project administration:** Andrew Kulmatiski, Janice Brahney, Michael C. Duniway.

**Resources:** Andrew Kulmatiski, Michael C. Duniway.

**Supervision:** Andrew Kulmatiski, Janice Brahney, Michael C. Duniway.

**Validation:** Andrew Kulmatiski, Michael C. Duniway.

**Writing – original draft:** Andrew Kulmatiski, Mehmet Ozturk.

**Writing – review & editing:** Andrew Kulmatiski, Janice Brahney, Michael C. Duniway.

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
