## [Decision Letter · Decision Letter 0]

21 Jun 2025

PONE-D-25-20429Season, wind speed, and seasonal rain are major drivers of a regional sediment transport modelPLOS ONE

Dear Dr. Kulmatiski,

Thank you for submitting your manuscript to PLOS ONE. After careful consideration, we feel that it has merit but does not fully meet PLOS ONE’s publication criteria as it currently stands. Therefore, we invite you to submit a revised version of the manuscript that addresses the points raised during the review process.

Dear Authors,

We have received two reviews of your manuscript, both highlighting minor revisions to be made. The primary concern raised by the reviewers is the need for more detailed explanations in the Methods section.

Additionally, we encourage you to strengthen your argument in the Discussion and Introduction sections, particularly in emphasizing the novelty of your contribution.

Please ensure that all necessary corrections requested by the reviewers are incorporated into your revised manuscript.

We look forward to receiving your updated submission.

Sincerely,

Academic Editor

We look forward to receiving your revised manuscript.

Kind regards,

Jamil Alexandre Ayach Anache

Academic Editor

PLOS ONE

**Journal Requirements:**

1. When submitting your revision, we need you to address these additional requirements. Please ensure that your manuscript meets PLOS ONE's style requirements, including those for file naming. The PLOS ONE style templates can be found at https://journals.plos.org/plosone/s/file?id=wjVg/PLOSOne_formatting_sample_main_body.pdf and https://journals.plos.org/plosone/s/file?id=ba62/PLOSOne_formatting_sample_title_authors_affiliations.pdf 2. In your Methods section, please provide additional information regarding the permits you obtained for the work. Please ensure you have included the full name of the authority that approved the field site access and, if no permits were required, a brief statement explaining why. 3. Please note that PLOS ONE has specific guidelines on code sharing for submissions in which author-generated code underpins the findings in the manuscript. In these cases, we expect all author-generated code to be made available without restrictions upon publication of the work. Please review our guidelines at https://journals.plos.org/plosone/s/materials-and-software-sharing#loc-sharing-code and ensure that your code is shared in a way that follows best practice and facilitates reproducibility and reuse. 4. Thank you for stating in your Funding Statement: This research was supported by the Utah Agricultural Experiment Station, Utah State University, and approved as journal paper number 9854. The Ministry of National Education and the Ministry of Agriculture and Forest of Turkey supported M. Ozturk.   Please provide an amended statement that declares *all* the funding or sources of support (whether external or internal to your organization) received during this study, as detailed online in our guide for authors at http://journals.plos.org/plosone/s/submit-now.  Please also include the statement “There was no additional external funding received for this study.” in your updated Funding Statement. Please include your amended Funding Statement within your cover letter. We will change the online submission form on your behalf. 5. We noted in your submission details that a portion of your manuscript may have been presented or published elsewhere. “Datasets used in the research were published separately, but not analyzed. Tyree, G.L., L. A. Zeller, J. Belnap, E. L. Geiger, T. W. Nauman, and M. C. Duniway. 2024, “Dust mass and horizontal aeolian sediment flux data from a sampler network on the Colorado Plateau, USA.” U.S. Geological Survey data release. https://doi.org/10.5066/P1ZHQX9W”.Please clarify whether this [conference proceeding or publication] was peer-reviewed and formally published. If this work was previously peer-reviewed and published, in the cover letter please provide the reason that this work does not constitute dual publication and should be included in the current manuscript. 6. Please include your full ethics statement in the ‘Methods’ section of your manuscript file. In your statement, please include the full name of the IRB or ethics committee who approved or waived your study, as well as whether or not you obtained informed written or verbal consent. If consent was waived for your study, please include this information in your statement as well. 7. We note that Figure 1 in your submission contain satellite images which may be copyrighted. All PLOS content is published under the Creative Commons Attribution License (CC BY 4.0), which means that the manuscript, images, and Supporting Information files will be freely available online, and any third party is permitted to access, download, copy, distribute, and use these materials in any way, even commercially, with proper attribution. For these reasons, we cannot publish previously copyrighted maps or satellite images created using proprietary data, such as Google software (Google Maps, Street View, and Earth). For more information, see our copyright guidelines: http://journals.plos.org/plosone/s/licenses-and-copyright. We require you to either present written permission from the copyright holder to publish these figures specifically under the CC BY 4.0 license, or remove the figures from your submission: a. You may seek permission from the original copyright holder of Figure 1 to publish the content specifically under the CC BY 4.0 license.   We recommend that you contact the original copyright holder with the Content Permission Form (http://journals.plos.org/plosone/s/file?id=7c09/content-permission-form.pdf) and the following text:“I request permission for the open-access journal PLOS ONE to publish XXX under the Creative Commons Attribution License (CCAL) CC BY 4.0 (http://creativecommons.org/licenses/by/4.0/). Please be aware that this license allows unrestricted use and distribution, even commercially, by third parties. Please reply and provide explicit written permission to publish XXX under a CC BY license and complete the attached form.” Please upload the completed Content Permission Form or other proof of granted permissions as an "Other" file with your submission. In the figure caption of the copyrighted figure, please include the following text: “Reprinted from [ref] under a CC BY license, with permission from [name of publisher], original copyright [original copyright year].” b. If you are unable to obtain permission from the original copyright holder to publish these figures under the CC BY 4.0 license or if the copyright holder’s requirements are incompatible with the CC BY 4.0 license, please either i) remove the figure or ii) supply a replacement figure that complies with the CC BY 4.0 license. Please check copyright information on all replacement figures and update the figure caption with source information. If applicable, please specify in the figure caption text when a figure is similar but not identical to the original image and is therefore for illustrative purposes only.The following resources for replacing copyrighted map figures may be helpful: USGS National Map Viewer (public domain): http://viewer.nationalmap.gov/viewer/The Gateway to Astronaut Photography of Earth (public domain): http://eol.jsc.nasa.gov/sseop/clickmap/Maps at the CIA (public domain): https://www.cia.gov/library/publications/the-world-factbook/index.html and https://www.cia.gov/library/publications/cia-maps-publications/index.htmlNASA Earth Observatory (public domain): http://earthobservatory.nasa.gov/Landsat: http://landsat.visibleearth.nasa.gov/USGS EROS (Earth Resources Observatory and Science (EROS) Center) (public domain): http://eros.usgs.gov/#Natural Earth (public domain): http://www.naturalearthdata.com/

**Additional Editor Comments:**

Dear Authors,

We have received two reviews of your manuscript, both highlighting minor revisions to be made. The primary concern raised by the reviewers is the need for more detailed explanations in the Methods section.

Additionally, we encourage you to strengthen your argument in the Discussion and Introduction sections, particularly in emphasizing the novelty of your contribution.

Please ensure that all necessary corrections requested by the reviewers are incorporated into your revised manuscript.

We look forward to receiving your updated submission.

Sincerely,

Academic Editor

Reviewers' comments:

Reviewer's Responses to Questions

**Comments to the Author**

1. Is the manuscript technically sound, and do the data support the conclusions?

Reviewer #1: Yes

Reviewer #2: Yes

2. Has the statistical analysis been performed appropriately and rigorously? 

Reviewer #1: Yes

Reviewer #2: Yes

3. Have the authors made all data underlying the findings in their manuscript fully available?

Reviewer #1: Yes

Reviewer #2: Yes

4. Is the manuscript presented in an intelligible fashion and written in standard English?

Reviewer #1: Yes

Reviewer #2: Yes

5. Review Comments to the Author

**Reviewer #1:**  Summary: in this study, the authors tested several statistical models to explain the variance observed in a dataset of aeolian (horizontal) sediment fluxes measured from 2004 to 2018 in a network of 52 masts of sand traps located on the Colorado Plateau (USA). They showed that the use of a random forest statistical model instead of a regression tree allowed explaining 91% of the variance in sediment transport vs. 58% before. Increasing sample size (adding 4 years of measurements) did not improve the results so does adding information on soil moisture, grazing condition or soil roughness. The random forest model developed by the authors allow classifying the parameters that explain the variance observed in the dataset with wind speed being the primary determinant of sediment transport, followed by soil moisture.

Major comment:

The study proposed by Kulmatiski et al. is of interest because it discusses the interest of using random forest statistical models to explain the variance observed in large dataset and gives insight on how to optimize its use. It also gives advice on the requirements of future networks to analyse aeolian sediment transport at landscape scale. The manuscript is well organized and written. All findings are supported by the results presented in the study. However, some clarifications in the Materials and Methods section are needed.

For all these reasons, I recommend to consider the publication of the present manuscript submitted to PLOS One after questions listed below have been addressed.

• Minor comments:

Title and running title should be revised to precise that sediment transport concerns only wind erosion.

ABSTRACT

l. 42-44: Not sure to understand the sentence. The word “and” should be removed, shouldn’t it? Please revise.

INTRODUCTION

l. 62-65: the authors are imprecise. Wind speed is not an aerodynamic force. Here, the authors must refer to drag or lift. In the same way, soil moisture, soil crusts and vegetation structures are not forces that oppose soil particle removal. Here, the authors must refer to cohesive forces such as capillary forces or chemical binding forces. This sentence must be revised.

MATERIALS AND METHODS

2.2 Sediment Transport Monitoring

l. 144-145: the authors chose to use two levels, 50 cm and 100 cm. In Flagg et al. (2014), it is stated that “We report sediment flux additively as the sum of sediment flux at 50 and 100 cm.” Is it the same here? Please specify.

l. 149-151: in Flagg et al. (2014), it is stated that “Sediment samples were sometimes collected with de-ionized water if the traps were filled with water, and later freeze dried to remove the water. Samples were weighed to within 0.001 g (Denver Instruments Microscale, Denver, CO)”. Here, another protocol was used. Please explain why or correct.

2.5 Soil Moisture Modeling

l. 191-192: why did the authors assign a value of 50 cm for plant height and a value of 1 for leaf area index. Please explain.

2.5 Data Analyses

Section number is incorrect and must be changed to 2.6

l. 212: the authors should refer to Table 1. Please verify.

l. 224: please define VIMP.

RESULTS

3.1 Sediment Transport

l. 241: why did the authors average data at 50 cm and 100 cm whereas Flagg et al. (2014) summed the values? Please explain.

l. 242: what is the interest to present the averaged value of sediment transport on 11 years and 52 sites considering the large scales of variations both in time (meteorological conditions) and space (different landscapes)? Please explain.

3.3 Random forest and regression tree models

Section number is incorrect and must be changed to 3.2

l. 273-274: the authors state from Figure 6 that “sediment flux increased as soil moisture at 30 cm increased.” Please, elaborate: give numbers, is it significant?

DISCUSSION

l. 300-305: the authors should temper their words as the model they had designed is specific to the region where it was developed. Please, rephrase.

REFERENCES

l. 408-411: this reference is misplaced and must be moved after Ballatyne et al. (2011).

l. 419-421: Belnap et al. (2009) is not cited in the text. To be removed. On the other hand, Belnap et al. (1997) cited l. 365 is missing and must be added in this section.

l. 446-447: Crawley and Nickling (2003) is not cited in the text. To be removed. On the other hand, Crawley et al. (2003) cited l. 342 is missing and must be added in this section.

Flagg et al. (2023) cited l. 84 is missing and must be added in this section.

l. 461-462: Floyd and Gill (2011) is not cited in the text. To be removed.

Fryrear (1986) cited l. 132-133 is missing and must be added in this section.

l. 468-469: Gillette (1979) is not cited in the text. To be removed. On the other hand, Gillette et al. (1997) cited l. 75 is missing and must be added in this section.

l. 474-476: Goldstein et al. (2010) is not cited in the text. To be removed.

l. 488-491: this reference is misplaced and must be moved after Hoover et al. (2021).

l. 492-494: please check the spelling of Hoffmann, written Hoffman when cited l. 102.

l. 510-517: Lemos and Lutz (2010), Li et al. (2007), and Li et al. (2013) are not cited in the text. To be removed.

Miller et al. (2006) cited l. 160 is missing and must be added in this section.

l. 546-548: Saha (2003) is not cited in the text. To be removed.

Simunek et al. (2008) cited l. 181 is missing and must be added in this section.

l. 558-559: Tchakerian (2014) is not cited in the text. To be removed.

l. 572-573: Webb & Strong (2011) and Webb et al. (2019) are not cited in the text. To be removed. On the other hand, Webb et al. (2011) cited l. 60 is missing and must be added in this section.

FIGURES

Figures 2 to 4: please add ticks on the y-axis for clarity.

- Some typos:

l. 160: replace “eta al.” by “et al.”.

l. 299: replace “Bruiec” by “Brieuc”.

**Reviewer #2:**  The authors used 52 horizontal sediment flux and tested several approaches to compute sediment transport induced by wind and correlated with other variables such as soil moisture and land cover. They used random forest statistical analysis to improve results and made comparisons against a previous study using a regression tree.

The authors presented a good introduction, establishing what limitations they wanted to overcome and what goals they wanted to achieve. As a continuation of previous works, the authors have given the correct credits, but sometimes they forgot to provide more details of the methods. The results are consistent with the methods, but the presentation can be improved. Also, in the discussion part, more detais are needed to clarify the results.

Comments:

• Line 109: Include the reference to Fig 1 after “study area”,

• Line 122: Remove the link

• Lines 191-192: You need to explain why you made those assumptions.

• Line 198: “(F1,3813 = 3382, p < 0.05, R2 = 0.47).” what are these variables and what do they mean?

• Line 202: “While we primarily rely on random forest analyses”… It is important to give some brief explanation about these methods cited in this paragraph.

• Line 242: “of 6.96 ± 26.09”... What happens when we have 6.96 – 26.09? is it possible? What means this large SD?

• Line 247: 3.1 and 3.3, what about 3.2?

• Lines 262-265: what is the log sum? what is the partial dependence plots?

• Lines 327-332: References! Is it possible that when soil moisture at 30cm is increasing the soil moisture at 5cm is decreasing? Do you also reflect that simulated soil moisture presented a low R2 and the uncertainty is higher than the correlation between sediment transport and soil moisture?

• Figures 2-4: What is A, B, C D, etc.?

• Figure 6: I did not understand it. Figure 6 - wind (could be 'a') peak occurs for 2.5. You need to explain what are the axis putting units. Overall, the figures and their captions are not self-explanatory.

6. PLOS authors have the option to publish the peer review history of their article (what does this mean? ). If published, this will include your full peer review and any attached files.

**Do you want your identity to be public for this peer review?** For information about this choice, including consent withdrawal, please see our Privacy Policy .

Reviewer #1: No

Reviewer #2: No

---

## [Author Response · Author response to Decision Letter 1]

25 Jul 2025

Response to reviewer comments

Please find point-by-point responses below.

PONE-D-25-20429

Season, wind speed, and seasonal rain are major drivers of a regional sediment transport model

PLOS ONE

Dear Dr. Kulmatiski,

Thank you for submitting your manuscript to PLOS ONE. After careful consideration, we feel that it has merit but does not fully meet PLOS ONE’s publication criteria as it currently stands. Therefore, we invite you to submit a revised version of the manuscript that addresses the points raised during the review process.

Dear Authors,

We have received two reviews of your manuscript, both highlighting minor revisions to be made. The primary concern raised by the reviewers is the need for more detailed explanations in the Methods section.

Additionally, we encourage you to strengthen your argument in the Discussion and Introduction sections, particularly in emphasizing the novelty of your contribution.

Please ensure that all necessary corrections requested by the reviewers are incorporated into your revised manuscript.

We look forward to receiving your updated submission.

Sincerely,

Academic Editor

Thanks for this opportunity to revise our manuscript. You will find point-by-point responses below. As suggested, we have also strengthened the Introduction and Discussion by highlighting the novelty of this paper.

We look forward to receiving your revised manuscript.

Kind regards,

Jamil Alexandre Ayach Anache

Academic Editor

PLOS ONE

Journal Requirements:

Response: Thanks, we have reformatted the manuscript to meet PLOS ONE’s style requirements.

Response: We have added the following to the methods: ‘This study involved collecting dust samples from an established network of dust collectors placed on public lands. No permits or sampling ethics approvals were needed for this work.’

Response: We have included our code as appendices in the revision.

This research was supported by the Utah Agricultural Experiment Station, Utah State University, and approved as journal paper number 9854. The Ministry of National Education and the Ministry of Agriculture and Forest of Turkey supported M. Ozturk.

Response: We have amended our Funding Statement as suggested and attached the revised statement to our cover letter.

5. We noted in your submission details that a portion of your manuscript may have been presented or published elsewhere. “Datasets used in the research were published separately, but not analyzed. Tyree, G.L., L. A. Zeller, J. Belnap, E. L. Geiger, T. W. Nauman, and M. C. Duniway. 2024, “Dust mass and horizontal aeolian sediment flux data from a sampler network on the Colorado Plateau, USA.” U.S. Geological Survey data release. https://doi.org/10.5066/P1ZHQX9W”.

Response: The dataset was published by the US Geological Survey, but was not peer-reviewed or analyzed. Simply the raw data was published. We have addressed this point in our cover letter.

Response: We have added the following to the Methods: ‘This study involved collecting dust samples from an established network of dust collectors placed on public lands. No permits or sampling ethics approvals were needed for this work.’

7. We note that Figure 1 in your submission contain satellite images which may be copyrighted. All PLOS content is published under the Creative Commons Attribution License (CC BY 4.0), which means that the manuscript, images, and Supporting Information files will be freely available online, and any third party is permitted to access, download, copy, distribute, and use these materials in any way, even commercially, with proper attribution. For these reasons, we cannot publish previously copyrighted maps or satellite images created using proprietary data, such as Google software (Google Maps, Street View, and Earth). For more information, see our copyright guidelines: http://journals.plos.org/plosone/s/licenses-and-copyright.

We require you to either present written permission from the copyright holder to publish these figures specifically under the CC BY 4.0 license, or remove the figures from your submission:

Response: We have changed the figure and now include a publicly available USGS map.

Response: We have reformatted the References to meet Plos One style requirements. We are aware of no papers that have been retracted.

Additional Editor Comments:

Dear Authors,

We have received two reviews of your manuscript, both highlighting minor revisions to be made. The primary concern raised by the reviewers is the need for more detailed explanations in the Methods section.

Additionally, we encourage you to strengthen your argument in the Discussion and Introduction sections, particularly in emphasizing the novelty of your contribution.

Please ensure that all necessary corrections requested by the reviewers are incorporated into your revised manuscript.

We look forward to receiving your updated submission.

Sincerely,

Academic Editor

Response: Thanks for this opportunity to revise our manuscript. We have added the following comments to the Introduction and Discussion to clarify the novel contribution of this manuscript:

In the Introduction we have added the following: ‘Soil moisture, therefore, is known to be important at small scales and is likely to be important at large scales, but we are not aware of a study that has assessed the effect of soil moisture on sediment transport at large scales. Here, we address this knowledge gap by including estimates of soil moisture as a predictor of sediment flux in regional sediment flux measurement network.’

In the Discussion, we have added the following:’ This study improved upon previous analyses of a regional dust flux dataset by 1) testing the effects of soil moisture, 2) greater dataset size, 3) soil roughness measurements, and by using a random forest (RF) modeling approach, which was previously found to improve upon previous regression tree modeling approaches.’

Reviewers' comments:

Reviewer's Responses to Questions

Comments to the Author

1. Is the manuscript technically sound, and do the data support the conclusions?

Reviewer #1: Yes

Reviewer #2: Yes

2. Has the statistical analysis been performed appropriately and rigorously?

Reviewer #1: Yes

Reviewer #2: Yes

3. Have the authors made all data underlying the findings in their manuscript fully available?

Reviewer #1: Yes

Reviewer #2: Yes

4. Is the manuscript presented in an intelligible fashion and written in standard English?

Reviewer #1: Yes

Reviewer #2: Yes

5. Review Comments to the Author

Reviewer #1: Summary: in this study, the authors tested several statistical models to explain the variance observed in a dataset of aeolian (horizontal) sediment fluxes measured from 2004 to 2018 in a network of 52 masts of sand traps located on the Colorado Plateau (USA). They showed that the use of a random forest statistical model instead of a regression tree allowed explaining 91% of the variance in sediment transport vs. 58% before. Increasing sample size (adding 4 years of measurements) did not improve the results so does adding information on soil moisture, grazing condition or soil roughness. The random fo

---

## [Decision Letter · Decision Letter 1]

1 Sep 2025

PONE-D-25-20429R1Season, wind speed, and seasonal rain are major drivers of a regional aeolian sediment transport modelPLOS ONE

Dear Dr. Kulmatiski,

Thank you for submitting your manuscript to PLOS ONE. After careful consideration, we feel that it has merit but does not fully meet PLOS ONE’s publication criteria as it currently stands. Therefore, we invite you to submit a revised version of the manuscript that addresses the points raised during the review process.

We look forward to receiving your revised manuscript.

Kind regards,

Jamil Alexandre Ayach Anache

Academic Editor

PLOS ONE

Journal Requirements:

Additional Editor Comments:

Please, check Reviewer #1 final comments.

Reviewers' comments:

Reviewer's Responses to Questions

**Comments to the Author**

1. If the authors have adequately addressed your comments raised in a previous round of review and you feel that this manuscript is now acceptable for publication, you may indicate that here to bypass the “Comments to the Author” section, enter your conflict of interest statement in the “Confidential to Editor” section, and submit your "Accept" recommendation.

Reviewer #1: (No Response)

Reviewer #2: All comments have been addressed

2. Is the manuscript technically sound, and do the data support the conclusions?

Reviewer #1: Yes

Reviewer #2: Yes

3. Has the statistical analysis been performed appropriately and rigorously? 

Reviewer #1: Yes

Reviewer #2: Yes

4. Have the authors made all data underlying the findings in their manuscript fully available?

Reviewer #1: Yes

Reviewer #2: Yes

5. Is the manuscript presented in an intelligible fashion and written in standard English?

Reviewer #1: Yes

Reviewer #2: Yes

6. Review Comments to the Author

Reviewer #1: I thank the authors who answered all my comments convincingly.

However, there is still a mistake in the manuscript in l. 146 and in l. 248. Indeed, in Flagg et al. (2014), it is stated that “We report sediment flux additively as the sum of sediment flux at 50 and 100 cm.” In the answer the authors made to one of my question in the first round of review, they confim that they used the sum of 50 and 100 cm measurements, as in Flagg et al. (2014). Unfortunately, in l. 146 and l. 248, it is still written that the authors used the average of sediment transport from 50 and 100 cm heights. This must be corrected before the manuscript is suitable for publication in PLOS One.

Reviewer #2: (No Response)

7. PLOS authors have the option to publish the peer review history of their article (what does this mean? ). If published, this will include your full peer review and any attached files.

**Do you want your identity to be public for this peer review?** For information about this choice, including consent withdrawal, please see our Privacy Policy .

Reviewer #1: No

Reviewer #2: No

---

## [Author Response · Author response to Decision Letter 2]

8 Sep 2025

The only comment requiring attention was the following:

Reviewer #1: I thank the authors who answered all my comments convincingly.

However, there is still a mistake in the manuscript in l. 146 and in l. 248. Indeed, in Flagg et al. (2014), it is stated that “We report sediment flux additively as the sum of sediment flux at 50 and 100 cm.” In the answer the authors made to one of my question in the first round of review, they confim that they used the sum of 50 and 100 cm measurements, as in Flagg et al. (2014). Unfortunately, in l. 146 and l. 248, it is still written that the authors used the average of sediment transport from 50 and 100 cm heights. This must be corrected before the manuscript is suitable for publication in PLOS One.

Response: Thanks for catching this oversight. We have clarified that the sum was calculated and reported at lines 146 (155) and 248 (246)

---

## [Decision Letter · Decision Letter 2]

11 Sep 2025

Season, wind speed, and seasonal rain are major drivers of a regional aeolian sediment transport model

PONE-D-25-20429R2

Dear Dr. Kulmatiski,

We’re pleased to inform you that your manuscript has been judged scientifically suitable for publication and will be formally accepted for publication once it meets all outstanding technical requirements.

Kind regards,

Jamil Alexandre Ayach Anache

Academic Editor

PLOS ONE

Additional Editor Comments (optional):

Reviewers' comments:

Reviewer's Responses to Questions

**Comments to the Author**

1. If the authors have adequately addressed your comments raised in a previous round of review and you feel that this manuscript is now acceptable for publication, you may indicate that here to bypass the “Comments to the Author” section, enter your conflict of interest statement in the “Confidential to Editor” section, and submit your "Accept" recommendation.

Reviewer #1: All comments have been addressed

2. Is the manuscript technically sound, and do the data support the conclusions?

Reviewer #1: Yes

3. Has the statistical analysis been performed appropriately and rigorously? 

Reviewer #1: Yes

4. Have the authors made all data underlying the findings in their manuscript fully available?

Reviewer #1: Yes

5. Is the manuscript presented in an intelligible fashion and written in standard English?

Reviewer #1: Yes

6. Review Comments to the Author

Reviewer #1: (No Response)

7. PLOS authors have the option to publish the peer review history of their article (what does this mean? ). If published, this will include your full peer review and any attached files.

**Do you want your identity to be public for this peer review?** For information about this choice, including consent withdrawal, please see our Privacy Policy .

Reviewer #1: No

---

## [Editor Report · Acceptance letter]

PONE-D-25-20429R2

PLOS ONE

Dear Dr. Kulmatiski,

I'm pleased to inform you that your manuscript has been deemed suitable for publication in PLOS ONE. Congratulations! Your manuscript is now being handed over to our production team.

Kind regards,

on behalf of

Dr. Jamil Alexandre Ayach Anache

Academic Editor

PLOS ONE